# Coxsackievirus Group B3 Has Oncolytic Activity against Colon Cancer through Gasdermin E-Mediated Pyroptosis

**DOI:** 10.3390/cancers14246206

**Published:** 2022-12-15

**Authors:** Yejia Zhang, Tian Xu, Huizhen Tian, Jianfeng Wu, Xiaomin Yu, Lingbing Zeng, Fadi Liu, Qiong Liu, Xiaotian Huang

**Affiliations:** 1Department of Medical Microbiology, School of Medicine, Jiangxi Medical College, Nanchang University, Nanchang 330006, China; 2Department of Clinical Laboratory, The First Affiliated Hospital of Nanchang University, Nanchang 330006, China; 3The Department of Clinical Laboratory, Children’s Hospital of Jiangxi Province, Nanchang 330006, China

**Keywords:** coxsackievirus group B3, oncolytic viruses, colon cancer, caspase-3, GSDME, pyroptosis

## Abstract

**Simple Summary:**

Colon cancer is a common malignant tumor that occurs in the digestive tract of the colon. At present, the treatment of colon cancer is unsatisfactory, and its recurrence and drug resistance are common. It is important to find an effective treatment for colon cancer. In this study, we found that coxsackievirus group B3 can induce pyroptosis in colon cancer cell lines, which shows its oncolytic activity. Based on these findings, we believe that CVB3 may play an important role in the treatment of colon cancer as an oncolytic virus.

**Abstract:**

Colon cancer is the second leading cause of cancer-related death, and there are few effective therapies for colon cancer. This study explored the use of coxsackievirus group B3 (CVB3) as an oncolytic virus for the treatment of colon cancer. In this study, we verified that CVB3 induces death of colon cancer cell lines by directly observing cell morphology and Western blot results, and observed the oncolytic effects of CVB3 by constructing an immunodeficient nude mice model. Our data show that CVB3 induces pyroptosis in colon cancer cell lines. Mechanistically, we demonstrated that CVB3 causes cleavage of gasdermin E (GSDME), but not gasdermin D (GSDMD), by activating caspase-3. This leads to production of GSDME N-termini and the development of pores in the plasma membrane, inducing pyroptosis of colon cancer cell lines. We also demonstrate that CVB3-induced pyroptosis is promoted by reactive oxygen species (ROS). Finally, in vivo studies using immunodeficient nude mice revealed that intratumoral injection of CVB3 led to significant tumor regression. Our findings indicate that CVB3 has oncolytic activity in colon cancer cell lines via GSDME-mediated pyroptosis.

## 1. Introduction

Colon cancer (CC) is a malignant tumor of the colon, the third most common gastrointestinal cancer, and the third most common type of malignant tumor worldwide [1]. Although research on colon cancer is ongoing, it is still the second most common cause of death. Therefore, innovative methods of treating colon cancer are urgently required. Oncolytic viruses (OVs), a class of natural or modified viruses that have the ability to replicate in cancer cells and kill them, are a promising new approach to cancer therapy [2]. Because OVs have good security, OV-mediated cancer therapy has become an emerging immunotherapy method [2,3]. Many OVs have been studied as potential cancer treatments, including the herpes virus, adenovirus, coxsackievirus, reovirus, and adenovirus [4]. For example, T-VEC (genetically engineered HSV-1) has been approved by the FDA for the treatment of advanced melanoma [5]. 

Although developments have been made in OV therapy in recent years and despite it representing one of the latest strategies for treating cancer, only a small number of patients have benefited from OV therapy. Therefore, new OVs are required to play a better cancer treatment role. Coxsackievirus group B3 (CVB3) is a non-enveloped icosahedral RNA virus belonging to the genus Enterovirus of the family *Picornaviridae* [6]. It has been reported that CVB3 has oncolytic activity against a variety of cancer cells, such as breast cancer cells, endometrial cancer cells, lung cancer cells, and so on [7,8,9]. As a result, CVB3 is considered one of the most promising OVs for cancer therapy. Hazini et al. have confirmed that both wild-type CVB3 (WT-CVB3) and genetically engineered CVB3 have potential as oncolytic viruses in the treatment of colon cancer [10,11]. However, their work focused more on the effectiveness of cancer therapy while neglecting to explore the molecular mechanisms through which WT-CVB3 kills colon cancer cells. Therefore, we sought to clarify the molecular mechanisms of WT-CVB3-linked colon cancer cell death to better facilitate the clinical application of CVB3 as an oncolytic virus.

Pyroptosis is a newly discovered type of programmed cell death (PCD) that is distinct from apoptosis and necrosis and characterized by the formation of pores in the cell membrane, cell swelling and rupture, and the release of cell contents such as inflammatory cytokines, thus resulting in robust inflammatory responses [12,13]. Pyroptosis can be roughly divided into two pathways: the canonical gasdermin D (GSDMD) and caspase-1 (casp-1) pathway and a non-canonical GSDMD and casp-4/5/11-dependent non-canonical pathway [12]. However, recent studies have reported that casp-3, activation of which is usually associated with apoptosis, can cleave gasdermin E (GSDME) after casp-9 activation, and the resulting GSDME N-terminal fragment can be inserted into the cell membrane and form pores, thereby inducing pyroptosis [14].

Our lab is committed to the study of CVB3, which provides new potential treatment strategies for diseases such as myocarditis and pancreatitis [6,15]. Here, we report that wild-type CVB3 (WT-CVB3) can infect colon cancer cells lines in vivo and in vitro, causing cell death. Mechanistically, we demonstrate that CVB3 causes pyroptosis via the casp-3/GSDME pathway. In addition, using a xenograft model of colon cancer, we observed that tumor size decreased significantly after WT-CVB3 treatment. Our findings indicate that WT-CVB3 has a therapeutic effect against colon cancer via GSDME-mediated pyroptosis.

## 2. Materials and Methods

### 2.1. Ethics Statement

The live CVB3 strain was handled in a level 2 biosafety laboratory, and all experiments were performed in accordance with the institutional guidelines of Nanchang University. The animal protocol in this study was approved by the Animal Care and Use Committee of Nanchang University (NCU-1239). All animal experiments were performed in accordance with the Guidelines for the Care and Use of Laboratory Animals (Ministry of Health of China, 1998) and the guidelines of the Laboratory Animal Ethics Committee of Nanchang University.

### 2.2. Cell Culture, Virus, and Reagents

The human colon cancer cell line HT-29 was obtained from the State Key Laboratory of Food Science and Technology, Nanchang University (China), and the HCT-116 cell line was purchased from Hunan Fenghui Biotechnology Co., Ltd. (Changsha, China). Both cell lines were cultured in DMEM (Solarbio, Beijing, China) containing 10% fetal bovine serum (FBS, Every Green, Shanghai, China) at 37 °C in a 5% CO_2_ incubator. CVB3 (Nancy strain; GI: 323432) was propagated in HeLa cells and stored at −80 °C. All cells were infected with CVB3 at a multiplicity of infection (MOI) of 1. CVB3’s replication cycles were inconsistent in HCT116 and HT29 cells, leading us to select different experimental end-points to observe the phenomenon of virus-induced cell death and perform other related experiments. The caspase-3 inhibitor Z-DEVD-FMK (100 μM), ASK1 inhibitor selonsertib (100 μM), and the antioxidant N-acetyl-L-cysteine (NAC) (10 mM or 20 mM) were all purchased from MedChemExpress (Princeton, NJ, USA). Inhibitors were added to cells 2 h before the viruses. 

### 2.3. Fluorescence Inverted Microscope Imaging

After cells were infected with CVB3, cell morphology was observed under brightfield microscopy using an inverted fluorescence microscope (OLYMPUS, Tokyo, Japan) and representative images were taken. Cells were stained with 10 μg/mL of propidium iodide to visualize dead cells, and representative images were captured.

### 2.4. Flow Cytometry

Subsequently, 6 × 10^5^ human colon cancer cells were seeded in six-well plates and, after nearly 24 h of incubation, infected with CVB3. Cells (about 1 × 10^6^) were collected at indicated times and washed twice with cold PBS. Cells were stained using an Annexin V-FITC/PI Apoptosis Assay Kit (Biosharp, Beijing, China) according to the manufacturer’s instructions and were detected with flow cytometry using an Annexin V-FITC and PI-PE. Data were analyzed using FlowJo software (version 7.6) to obtain percentages of live and dead cells. Annexin-V bound to exposed phosphatidylserine in both apoptotic and pyroptotic cells.

### 2.5. LDH Release Assay

The level of lactate dehydrogenase (LDH) was detected using an LDH activity detection kit (Solarbio, China). The cells were plated and infected according to the method described above. After collecting the cells, LDH reagents were mixed according to the manufacturer’s protocol, added to the cell supernatants, and incubated for 3 min at room temperature. A microplate reader was used to measure optical density (OD) at 450 nm.

### 2.6. Cell Viability Assay

Cell viability was analyzed using a Cell Counting Kit-8 (CCK-8) assay (Biosharp, Beijing, China). A total of 7000 cells per well were seeded in a 96-well plate and, after nearly 24 h of incubation, infected with CVB3. After the indicated incubation times, 10 μL of CCK-8 solution was added per well and incubated for 1 h in a 5% CO_2_ incubator at 37 °C. A microplate reader was then used to measure OD at 450 nm.

### 2.7. Detection of Reactive Oxygen Species (ROS)

The level of ROS in cells was detected using diacetyldichlorofluorescein (DCFH-DA; Beyotime, Nantong, China). Cells were seeded in a 35 mm dish and were infected with CVB3 after approximately 24 h. After washing, the cells were stained with 10 μM of DCFH-DA in the dark for 30 min at 37 °C according to the manufacturer’s protocol, and green fluorescence was observed under a fluorescent inverted microscope after washing.

### 2.8. Western Blot

The cells were washed with PBS and lysed with lysis buffer (150 mM NaCl, 20 mM Tris HCl, 0.1% NP-40, pH 7.4). Equal amounts of protein were loaded onto 10% or 12% SDS-PAGE gels, separated, and then transferred to 0.45 μm polyvinylidene difluoride (PVDF) membranes (Millipore, Darmstadt, Germany). The membranes were blocked with 5% skimmed milk diluted by TBST for 2 h at room temperature, incubated with primary antibodies, including caspase-3/1 (1:1000, pAb; Proteintech, Chicago, IL, USA), caspase-9 (1:500, pAb; Proteintech, Chicago, IL, USA), PARP (1:1000, pAb; Proteintech, Chicago, IL, USA), GSDME (1:5000, pAb; Proteintech, Chicago, IL, USA), GSDMD (1:10,000, pAb; Immunoway, Plano, TX, USA), VP1 (1:1000, pAb; laboratory), GAPDH (1:50,000, mAb; Proteintech, USA), and β-actin (1:10,000, mAb; Proteintech, USA), at 4 °C overnight, and then incubated with secondary antibodies, either goat anti-rabbit IgG-HRP (1:5000, mAb; Boster, Pleasanton, CA, USA) or goat anti-mouse IgG-HRP (1:5000, mAb; Boster, USA), for about one hour at room temperature. HRP was visualized using an ECL Chemiluminescence Kit (Abbkine, Wuhan, China).

### 2.9. Nude Mouse Xenograft Assay

Male BALB/c nude mice were purchased from Hunan SJA Laboratory Animal Co., Ltd. (Changsha, China) and were given free access to food and water. HT-29 cells (3 × 10^6^) were injected subcutaneously into the armpits of 4-week-old mice. UV-inactivated CVB3 (UV-CVB3) and WT-CVB3 (5 × 10^6^ PFU) were intratumorally injected 24 days after inoculation, by which time the tumors had grown to approximately 100 mm^3^. The mice were weighed and tumor size was calculated every day. Tumor volume was calculated using the formula volume = S × L × L/2, where S and L are the short and long dimensions of the tumor, respectively. After the experiment, mice were sacrificed by cervical dislocation. After dissection, the tumor, pancreas, and heart tissue were split in two, with one half used for Western blotting and the other half for immunohistochemistry.

### 2.10. Immunohistochemical Analysis

Tumor tissues were fixed in 10% formalin and then embedded in paraffin and sliced. Slices were deparaffinized and rehydrated, with the slices then subjected to antigen retrieval before being washed three times with PBS (pH 7.4). After blocking, the tissue sections were incubated with anti-caspase-3 and anti-GSDME antibodies overnight at 4 °C, followed by incubation with a secondary antibody. After staining with 3,3′-diaminobenzidine (DAB), the slides were counterstained with hematoxylin, washed using tap water, and examined and imaged using a microscope (Nikon E100, Tokyo, Japan).

### 2.11. Statistical Analysis

All statistical analyses were performed using GraphPad Prism 5 software (GraphPad Software, San Diego, CA, USA). All experimental results were expressed as mean ± standard deviation. One- or two-way analysis of variance was performed, followed by Tukey’s *post hoc* test, to determine the significance of differences between the mean values of the experimental and control groups. *p* < 0.05 was considered statistically significant.

## 3. Results

### 3.1. CVB3 Induces Colon Cancer Cell Death

In order to determine whether CVB3 affects colon cancer cells, we infected two human colon cancer cell lines (HT-29 and HCT-116) with CVB3 and observed cell death in several different ways. Propidium iodide (PI) staining showed significant lytic cell death in HT-29 cells after CVB3 infection (Figure 1A). Next, we used an Annexin V-FITC/PI flow cytometry assay and CCK-8 assay to measure cytotoxicity in HT-29 cells. We found that CVB3 induced death in HT-29 cells, and that the number of dead cells increased as a function of infection time. Statistical analysis of the flow cytometry results revealed that more than 60% of the HT-29 cells died within 24 h of CVB3 infection (Figure 1B). These results suggest that CVB3 kills the HT-29 colon cancer cell line. We observed a similar effect on HCT-116 cells, with more than 90% of them dying within 48 h of CVB3 infection (Figure 1C,D).

### 3.2. CVB3 Induces Pyroptosis in Colon Cancer Cells and Causes LDH Release

Pyroptosis is an important natural immune response in the body and plays an important role in fighting infection. Hallmarks of pyroptosis are the release of LDH and the swelling of cells, which produces many bubble-like protrusions [16]. Using a microscope for direct morphological observation, HT-29 cells swelled after CVB3 infection, with characteristic large bubbles from the cell membrane (Figure 2A). CVB3 infection of HT-29 led to significant release of LDH compared to the control group, which increased as a function of infection time (Figure 2B). We observed swelling and LDH release in CVB3-infected HCT-116 cells similar to that observed in infected HT-29 cells (Figure 2C,D). These data suggest that CVB3 exerts its oncolytic activity by inducing pyroptosis.

### 3.3. CVB3 Induces Pyroptosis in Colon Cancer Cell Lines through the GSDME Pathway 

Since pyroptosis occurs in a variety of ways [12], we used Western blotting to gain insight into how pyroptosis is induced by CVB3. These results indicated that the degradation of GSDME was gradually increased as CVB3 infection time increased, and that PARP and casp-3/9 activation increased over time, indicating that CVB3-induced pyroptosis was dependent on GSDME activation (Figure 3A,B) [17]. We also investigated the canonical casp-1/GSDMD pyroptosis pathway. Neither casp-1 nor GSDMD were activated in HT-29 cells (Figure 3C and Appendix A). Although we observed the activation of the casp-1/GSDMD pathway in HCT116 cells 30 h after CVB3 infection (in comparison, GSDME-mediated pyroptosis occurred 12 h after CVB3 infection), we believe that CVB3 infection of HCT116 causes pyroptosis mainly via the casp-3/GSDME pathway, rather than the casp-1/GSDMD pathway that was subsequently activated (Figure 3D) [18]. The statistical analysis results of densitometry analysis of the digital images of Western blots in Figure 3 are shown in Appendix A. These data suggest that CVB3-induced pyroptosis of colon cancer cell lines occurs via the casp-3/GSDME pathway, rather than the casp-1/GSDMD pathway.

### 3.4. CVB3-Induced GSDME-Dependent Pyroptosis Is Promoted by ROS Signaling

Previous studies have reported that reactive oxygen species (ROS) can oxidize cardiolipin, leading to the release of cytochrome c and activation of casp-3, which ultimately causes pyroptosis by inducing the cleavage of GSDME [19]. ROS are thus the key molecules that induce GSDME-mediated pyroptosis. Therefore, we explored ROS levels after CVB3 infection of colon cancer cell lines. Under fluorescence microscopy, we detected DCFH-DA levels and observed a significant increase in ROS after infection with CVB3 (Figure 4A). In addition, we also validated this finding using NAC, an antioxidant [13]. Western blot analysis indicated that the ROS levels increased by CVB3 infection were significantly attenuated by NAC, and that NAC inhibited casp-3 activation and GSDME cleavage to some extent (Figure 4B). The statistical analysis results of densitometry analysis of the digital images of Western blots are shown in Appendix A. Similarly, significant green fluorescence was also observed after CVB3 infection of HCT-116 cells, and NAC also inhibited the cleavage of casp-3 and GSDME, suggesting elevated ROS levels. Western blotting indicated that NAC inhibited the activation of casp-3 and GSDME (Figure 4C,D and Appendix A).

In summary, these data suggest that CVB3 infection of colon cancer cells increases ROS levels, which drives CVB3-induced pyroptosis of colon cancer cell lines by activating the casp-3/GSDME pathway.

### 3.5. CVB3-Induced Pyroptosis Is Dependent on Casp-3-Mediated Cleavage of GSDME 

To further verify that CVB3-induced pyroptosis occurs through the casp-3/GSDME pathway but not the GSDMD pathway, we used two inhibitors to fully validate the pathway: Z-DEVD-FMK (an irreversible casp-3 inhibitor which significantly inhibits its activation) and selonsertib (an ASK1-specific inhibitor that inhibits phosphorylation of kinases downstream of ASK1) [20,21]. 

We incubated HT-29 cells with Z-DEVD-FMK for 2 h before removing the inhibitor and infecting the cells with CVB3. Z-DEVD-FMK-pretreated HT-29 cells exhibited reduced plasma membrane bubbling compared to cells infected with CVB3 only (Figure 5A). In addition, inhibitor pretreatment increased cell viability and reduced LDH release (Figure 5B,C). Western blotting results indicated that Z-DEVD-FMK inhibited the activation of casp-3, as expected, and that this inhibited the cleavage of PARP and GSDME (Figure 5D). Statistical analysis of densitometry analysis of the digital images of Western blots confirmed these results (Appendix A). Similarly, selonsertib decreased the number of pyroptotic cells, increased cell viability, decreased LDH release, and decreased CVB3-induced casp-3 and PARP activation (Figure 5E–H and Appendix A). 

These results demonstrate that inhibition of casp-3 activation can prevent the activation of GSDME and pyroptosis in HT-29 cells, and that the expression of VP1 (a structural protein of CVB3) is relatively low. This demonstrates that CVB3 induces pyroptosis of colon cancer cell lines through the casp-3/GSDME pathway and not the GSDMD pathway.

### 3.6. CVB3-Induced GSDME-Dependent Pyroptosis Inhibits the Growth of Colon Cancer Cells in Mice 

The above results demonstrate that CVB3 induces pyroptosis in vitro through the casp-3/GSDME pathway. To explore whether CVB3 has an anti-tumor effect in vivo, we injected HT-29 cells subcutaneously into BALB/c nude mice to make a xenograft model (Figure 6A). Tumor volume in mice treated with WT-CVB3 significantly decreased (Figure 6B), while tumor volume in mice injected with UV-CVB3 continued to increase (Figure 6C), which indicates that CVB3 can effectively kill colon cancer cells in vivo and that this effect is not related to the immune response. However, the overall body weight of mice injected with WT-CVB3 decreased, while that of mice injected with UV-CVB3 remained essentially unchanged (Figure 6D). This suggests that the WT-CVB3 treatment may have mildly damaged normal tissues and cells in the mice and not given mice a survival advantage. In order to determine whether the mechanism of tumor volume reduction we observed in vitro was casp-3/GSDME-mediated pyroptosis, we extracted proteins from the tumors and performed Western blot experiments. These showed that injection of CVB3 induced cleavage of casp-3/9, PARP, and GSDME (Figure 6E). The statistical analysis of densitometry analysis of these bands is shown in Appendix A. These results are consistent with the results of in vitro experiments. Immunohistochemical analysis demonstrated that intratumoral injection of WT-CVB3 led to greater activation of casp-3 and GSDME than UV-CVB3 (Figure 6F). Overall, intratumoral injection of CVB3 reduced tumor volume through casp-3/GSDME-mediated pyroptosis.

## 4. Discussions

Colon cancer is the third most common cancer in the world [22]. At present, the main treatments for colon cancer are surgical resection, immunotherapy, and chemotherapy [23]. However, incidence, recurrence, and metastasis rates are increasing [1,22]. OVs have the benefit of specifically targeting certain tumor cells, rather than just replicating cells [24], thereby reducing damage to healthy tissue. CVB3, despite causing pancreatitis and myocarditis, shows potential for treating colon cancer. Therefore, we explored the mechanism by which CVB3 exerts its oncolytic effect, laying the groundwork for its development into a better oncolytic virus.

Here, we report that CVB3 infects colon cancer cell lines, which reduces cell activities. The swelling of cells, accompanied by a pronounced extension of large bubbles from the plasma membrane and release of LDH, suggests that CVB3 induces pyroptosis. We explored the molecular mechanism by which CVB3 infection leads to pyroptosis in two colon cancer cell lines. Of note, after CVB3 infects HCT-116 cells, the casp-3/GSDME pathway is activated at 12 h post-inoculation, and the casp-1/GSDMD pathway is active at 30 h. This indicates that the pyroptosis caused by CVB3 action in HCT-116 cells may be mediated by the casp-3/GSDME pathway first, and that the casp-1/GSDMD pathway could then be activated by the action of cell metabolites and other substances. This phenomenon has also been reported in other studies, but the details of the regulatory mechanism need to be clarified with further research [25,26]. Based on the results with these two cell lines, we established that CVB3 infection is mainly mediated via the casp-3/GSDME pathway, leading to pyroptosis of colon tumor cells.

Studies have shown that ROS are associated with cancer development and cancer cell death [27,28]. In cancer cells, due to defects in oxidative metabolism, ROS accumulate [27]. At the same time, ROS are key molecules that induce GSDME-mediated pyroptosis, and their elevation stimulates the cleavage of casp-3 and GSDME in cancer cells, thereby inducing pyroptosis [29] (Figure 7). As an antioxidant, NAC inhibits casp-3 activation and GSDME cleavage to alleviate pyroptosis. This is consistent with our experimental results. Pretreatment with the casp-3 inhibitor Z-DEVD-FMK and the ASK1 inhibitor selonsertib inhibited CVB3-induced pyroptosis, indicating that CVB3-induced pyroptosis occurred through the casp-3/GSDME pathway. A mouse xenograft model demonstrated that CVB3 significantly reduced tumor volume in vivo, and that this was accompanied by cleavage of casp-3 and GSDME. These results suggest that CVB3 has the potential to be used to treat colon cancer. Of course, there are still some shortcomings in our study. The inhibition of ROS-promoted pyroptosis by NAC was not obvious in the Western blots. However, statistical analysis and microscopic observation of morphology did suggest that pretreatment with NAC inhibited ROS-driven pyroptosis.

Although our findings suggest that CVB3 has an oncolytic effect on colon cancer, mice injected with WT-CVB3 lost weight compared to mice injected with UV-CVB3. This suggests that intratumorally injected CVB3 may have spread, causing myocarditis or pancreatitis. This may have been because CVB3 can infect normal cells as well, which also express the coxsackievirus and adenovirus receptor (CAR) [30]. CVB3 infection is associated with severe illness in infants; however, in adults CVB3 generally induces mild self-limiting disease with flu-like symptoms [30]. Moreover, the immune system in the healthy human body is complete, so the side-effects caused by intratumoral injection of CVB3 may be mild [31]. In addition, as a single-stranded RNA virus, CVB3 replicates in the host cytosol without a DNA phase, resulting in an inability to integrate the viral genome into the host genome that minimizes the potential for genotoxicity [32]. CVB3 therefore has several therapeutic advantages. After the virus infects cells, it immediately induces robust death, and there are no oncogenes associated with the virus that might cause secondary tumorigenesis. The abnormal replication of tumor cells and the enrichment of some receptors on their surface create a microenvironment favoring viral replication that makes CVB3 better able to infect these hyperactive cells, causing apoptosis, pyroptosis, necrosis, or cell cycle arrest [33,34,35].

The inhibition of colon cancer tumor metastasis by oncolytic viruses is one of the key indicators of their therapeutic effect [1]. After the oncolytic virus infects cancer cells, it changes the tumor microenvironment, stimulating the antiviral components of the immune system which kill tumor cells. Activation of systemic immunity inhibits tumor spread and prevents recurrence [30,35]. Previous studies have also revealed the high potential efficacy of WT-CVB3 and genetically engineered CVB3 against metastatic colon cancer [10,11]. However, the molecular mechanism by which CVB3 acts against metastatic colon cancer has not been fully elucidated. Therefore, in subsequent experiments, we will mimic the metastasis of colon cancer by performing xenografts on both sides of immunodeficient nude mice, injecting the virus on only one side, determining the protein levels in signal pathways related to cell death, and measuring tumor volume on the opposite side to check for potential effects on metastatic tumors and regulatory mechanisms.

Intratumoral CVB3 administration is less preferable than intravenous injection of human patients in clinical trials. However, intravenous injection is more likely to aggravate the disease because CVB3 will infect normal cells and tissues. To prevent CVB3 from infecting healthy cells after entering the bloodstream, which can cause other diseases, WT-CVB3 should be modified to make it more suitable for clinical use. Its toxicity must be reduced and its safety and anti-tumor efficacy enhanced. The "hypoxia" feature is common in tumor cells but not seen in normal cells. We may be able to engineer CVB3 with a tumor-specific hypoxia-inducible factor (HIF)-responsive promoter which targets infected tumor cells without infecting normal cells [36]. In addition, we reported that the inhibition of pancreatic cell proliferation by CVB3 only required the D8 region of VP1 (a structural protein of CVB3) in a previous study [6]. Therefore, it may be possible that only VP1 or smaller fragments are needed to exert the oncolytic effect of CVB3. We have constructed a series of plasmids containing different VP1 fragments to determine the smallest fragment required for oncolytic effects. Furthermore, the CVB3 genome can be engineered to contain microRNAs behind pancreas- and heart-specific promoters to protect against pancreas and heart lesions, without affecting oncolytic activity [37,38]. Due to the high mutation rate of CVB3, the evolution of CVB3 can be directed by selecting beneficial mutations that improve anti-cancer efficacy [39]. Finally, we can also engineer tumor-toxic transgenes into CVB3 to further improve its anti-cancer efficacy [40,41]. In subsequent studies, we hope to combine the above improved methods to develop a truly effective, high-security oncolytic virus which can be used for clinical treatment.

Pyroptosis is a double-edged sword. It can open up new tumor treatment strategies by inducing cell death, but studies have also shown that excessive pyroptosis may also cause a serious inflammatory response, damage normal tissues, and lead to inflammatory diseases [42]. We confirmed that CVB3 infection will lead to colon tumor cell death by mediating pyroptosis; however, it also reduced the weight of mice. Therefore, in the subsequent clinical application of CVB3-mediated pyroptosis for colon tumor treatment, it is necessary to explore the dynamic balance between tumor cell death caused by pyroptosis and the inflammatory response induced by this process, as well as consider tumor cell targeting and the safety of viruses.

## 5. Conclusions

In summary, we demonstrate that CVB3 induced pyroptosis in colon cancer cell lines, that this was dependent on GSDME degradation downstream of casp-3, and that elevated ROS were required for casp-3/GSDME activation. Therefore, this study further confirmed the potential of CVB3 as an oncolytic virus for the treatment of colon cancer.

## Figures and Tables

**Figure 1 cancers-14-06206-f001:**
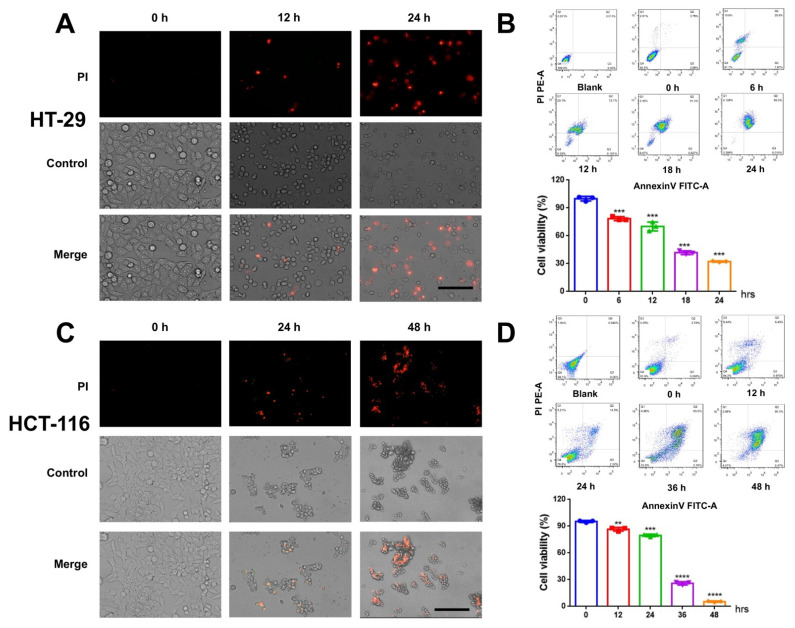
CVB3 induces HT-29 and HCT-116 cells death. HT-29 cells were infected with CVB3 (MOI = 1) for different periods of time. (**A**) Infected cells were observed using an inverted fluorescence microscope. Propidium iodide (PI) was added at 0, 12, and 24 h after HT-29 infection with CVB3. Magnification, ×200. Scale bar, 50 μm. (**B**) After CVB3 infection, HT-29 cells were collected at 0, 6, 12, 18, and 24 h and stained with Annexin V-FITC and PI for flow cytometry. Numbers represent the proportion of cells in each quadrant. Cell viability was determined using a CCK-8 assay. HCT-116 cells were infected with CVB3 (MOI = 1) for different periods of time. (**C**) Infected cells were observed using an inverted fluorescence microscope. Propidium iodide (PI) was added at 0, 24, and 48 h after HCT-116 infection with CVB3. Magnification, ×200. Scale bar, 50 μm. (**D**) After CVB3 infection, HCT-116 cells were collected at 0, 12, 24, 36, and 48 h and stained with Annexin V-FITC and PI for flow cytometry. Numbers represent the proportion of cells in each quadrant. Cell viability was determined using a CCK-8 assay. The data are expressed as the mean ± standard deviation of three independent experiments. ** *p* < 0.01, *** *p* < 0.001, **** *p* < 0.0001.

**Figure 2 cancers-14-06206-f002:**
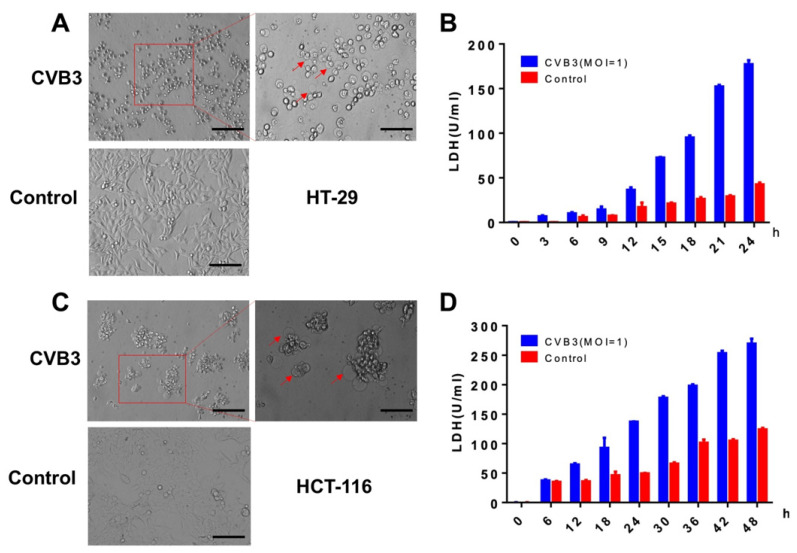
CVB3 induces colon cancer cell pyroptosis and leads to increased release of LDH. (**A**,**C**) Uninfected and CVB3-infected HT-29 and HCT-116 cells were imaged 24 h after infection. The red arrows point to bubbles (indicative of pyroptosis) that appeared from the plasma membrane. Left: magnification, ×100; scale bar, 100 μm. Right: magnification, ×200; scale bar, 50 μm. (**B**,**D**) Release of LDH from HT-29 and HCT-116 cells infected with CVB3.

**Figure 3 cancers-14-06206-f003:**
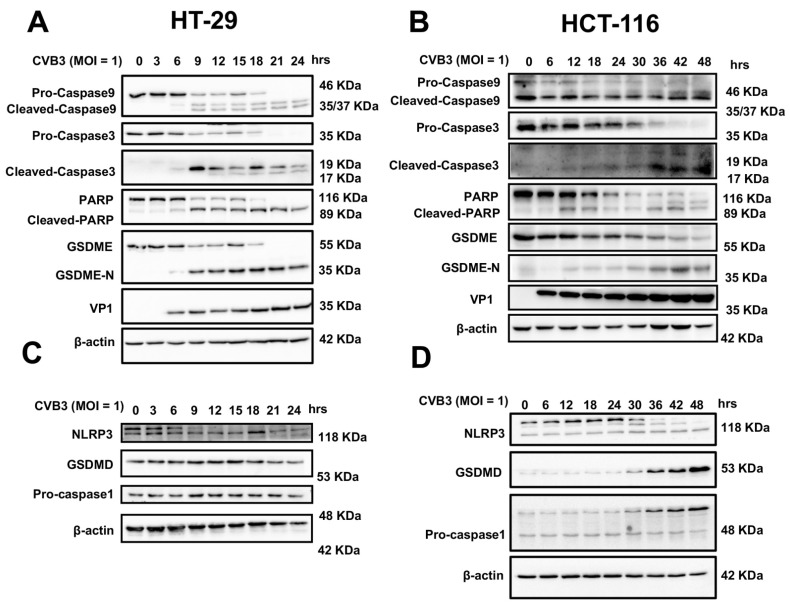
GSDME but not GSDMD is cleaved in CVB3-induced pyroptosis in colon cancer cells. (**A**,**B**) Western blots of GSDME and GSDME N terminus, PARP and cleaved PARP, and pro- and cleaved caspase-3/9 in HT-29 and HCT-116 cells infected with CVB3 for different periods of time. VP1 protein, as the structural protein of CVB3, can reflect the viral load in cells. (**C**,**D**) Western blots of GSDMD, NLRP3, and pro-caspase-1 in HT-29 and HCT-116 cells infected with CVB3 for different periods of time.

**Figure 4 cancers-14-06206-f004:**
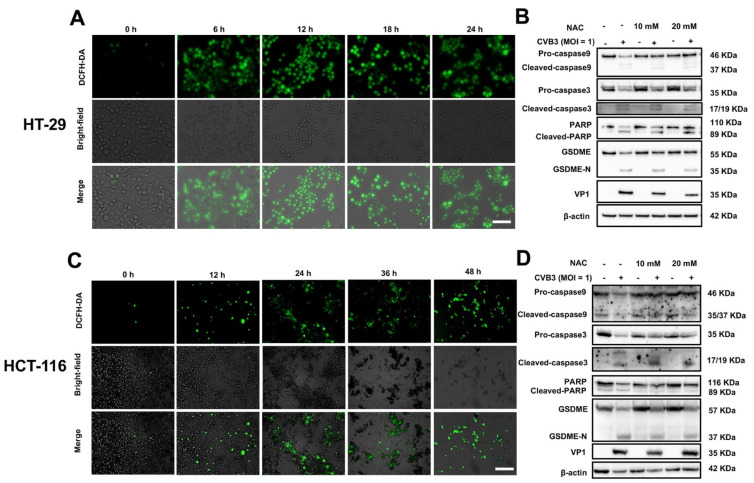
CVB3-induced pyroptosis is facilitated by ROS. (**A**,**C**) Brightfield and fluorescence images of CVB3-infected cells. ROS concentration in HT-29 and HCT-116 cells was measured using DCFH-HA, with green fluorescence indicating the presence of ROS. Magnification, ×100. Scale bar, 100 μm. (**B**,**D**) HT-29 and HCT-116 cells were pretreated with or without NAC (10 or 20 mM) for 2 h and then infected with CVB3 for 24 h, and cleavage of caspase-3 and GSDME was analyzed using Western blots.

**Figure 5 cancers-14-06206-f005:**
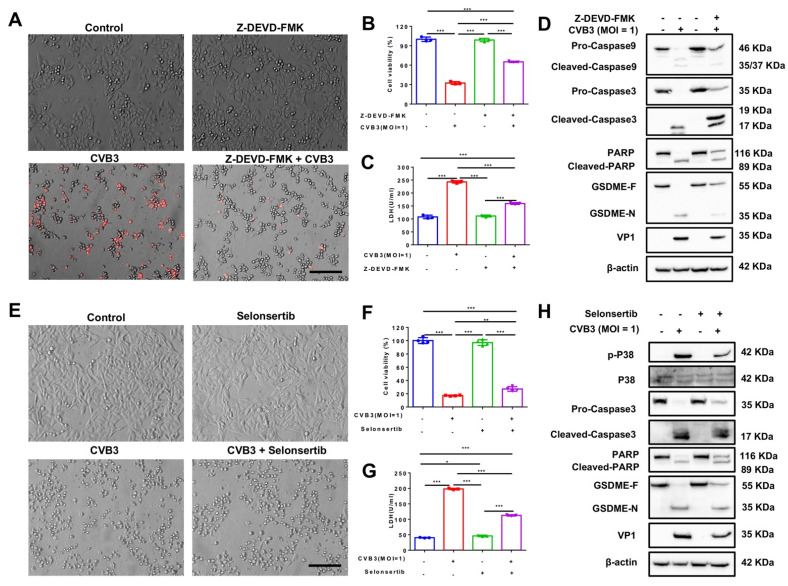
The caspase-3-specific inhibitor Z-DEVD-FMK and ASK1 inhibitor selonsertib inhibit the cleavage of GSDME. HT-29 cells were pretreated with or without Z-DEVD-FMK and selonsertib for 2 h and then infected with CVB3 for 24 h. (**A**,**E**) Representative brightfield images. Magnification, ×200. Scale bar, 50 μm. (**B**,**C**,**F**,**G**) CCK-8 and LDH assays for the detection of HT-29 cell viability and cytotoxicity after CVB3 infection. (**D**,**H**) Western blots to analyze GSDME and GSDME N terminus, PARP and cleaved PARP, and pro- and cleaved caspase-3/9 in HT-29 cells. * *p <* 0.05, ** *p <* 0.01, *** *p <* 0.001.

**Figure 6 cancers-14-06206-f006:**
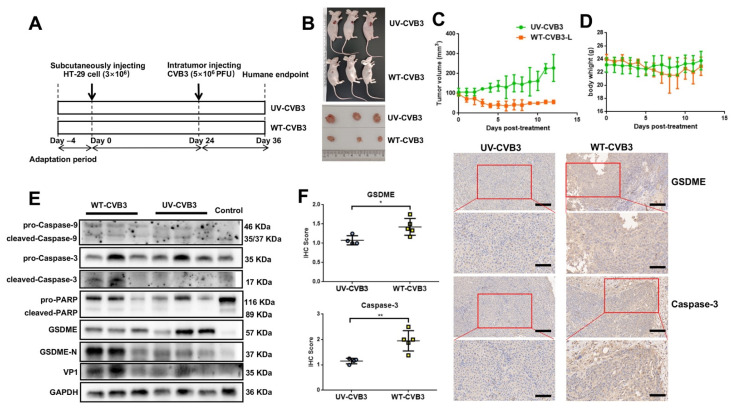
CVB3 induces pyroptosis of colon cancer cells in vivo. (**A**) Schematic of the in vivo experiment. Mice were first adapted to their new home cages, after which they were subcutaneously injected with 3 × 10^6^ HT-29 cells. After 24 days, when the tumors were approximately 100 mm^3^ in volume, WT-CVB3 or UV-CVB3 (5 × 10^6^ PFU) were injected intratumorally, and humane executions were performed after a further 12 days. (**B**) Representative images of tumors with intratumoral injection of UV-CVB3 or WT-CVB3 (n = 3) after 12 days. (**C**,**D**) Tumor volume and body weight over time. Data are represented as mean ± standard deviation (n = 3). Statistical analysis was performed using two-way ANOVA. (**E**) Western blot to analyze cleavage of caspase-9, caspase-3, PARP, and GSDME in tumor tissue treated with UV-CVB3 or WT-CVB3. (**F**) Immunohistochemical analysis of cleaved caspase-3 and GSDME N terminus in tumor sections. WT-CVB3 injection increased the expression of GSDME and caspase-3 in the tumor tissues of treated mice compared to the UV-CVB3 group. Representative images are shown (n = 3). Magnification, ×100. Scale bar, 100 μm (up), 40 μm (down). * *p <* 0.05, ** *p <* 0.01.

**Figure 7 cancers-14-06206-f007:**
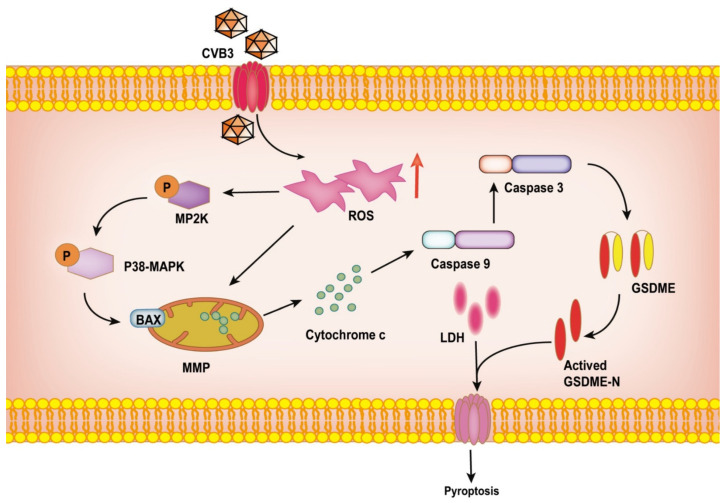
Schematic representation of the oncolytic effect of CVB3 in HT-29 cells. ROS are elevated by CVB3 infection of HT-29 cells, which promotes the release of cytochrome c. This activates the cleavage of caspase-9, which then activates caspase-3. Activated caspase-3 cleaves GSDME, leading to LDH release. The resulting GSDME N terminus has cell membrane pore-forming activity, which induces pyroptosis.

## Data Availability

The datasets used and/or analyzed during the current study are available from the corresponding author upon reasonable request.

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
