# Peer review of "Coxsackievirus Group B3 Has Oncolytic Activity against Colon Cancer through Gasdermin E-Mediated Pyroptosis"

_cancers, 2022, doi:10.3390/cancers14246206_

Round 1
Reviewer 1 Report
.
In their study, Zhang et al investigated the extent to which CVB3 can induce pyroptosis, a major mechanism leading to cell death, in colorectal carcinoma cells and in colorectal tumors in vivo. They demonstrated that CVB3 treatment induced cell death in vitro by pyroptosis via caspase-3-mediated cleavage of gasdermin E (GSDME). Moreover, CVB3-induced pyroptosis was promoted by reactive oxygen species (ROS). CVB3-induced pyroptosis in was confirmed in vivo after infection of colorectal tumors with CVB3.
The study is very well done and well written. Although it has already been shown that CVB3 can induce pyroptosis in CVB3-infected cells, this is the first time that this has been done in the context of oncolytic virotherapy in colorectal tumor cells. Thus, the focus of the paper and the main messages relate to elucidating the mechanism of pyroptosis occurring in colorectal tumor cells rather than the efficacy and safety or further development of CVB3 virotherapy.
Main criticism
It would certainly have been better if the authors had focused on elucidating the mechanism of pyroptosis by CVB3 in colorectal cells and colorectal carcinomas rather than repeatedly focusing on CVB3 virotherapy. This is particularly concerning because the presentations on CVB3 virotherapy give a rather distorted picture of the current state of the art. For example, there are already a number of papers that explicitly address the treatment of tumors, including colorectal carcinomas, with oncolytic CVB3. These (for example, Hazini et al. 2018 and 2021) have shown that wild-type CVB3 causes severe side effects and that more advanced CVB3s (CVB3 mutants or microRNA-regulated CVB3s) are more suitable for cancer therapy than the wild-type used in this study. Unfortunately, these studies were not cited and the data generated by the authors were not presented in the context of the previous results. Another example from the discussion section should be mentioned. The authors mention that in the future they plan to investigate the potential of CVB3 in the treatment of metastatic colorectal cancer by inducing two tumors in nude mice, one of which is injected with a CVB3. These studies have already been done in the 2018 and 2021 papers by Hazini et al. The introduction and the discussion should therefore be carefully revised once again.
Minor points.
1) In Figure 2C, arrows should be added (as in 2B) to show the pyroptosis bubble.
2 In Figure 3C, the expression of GSDMD is shown in the Western blot. Several bands can be seen here. Which of the bands corresponds to GSDMD? Figure 3C also shows an increase in the second band from the top for GSDMD starting at 15 hours and a decrease in the top band for pro-caspase-1. The significance of the bands and the reason for the change is not clear to the reviewer and is not explained in the text but should be explained.
3, Figure 6F. The figure is much too small; even at maximum magnification on the computer, the differences are barely visible. A sentence should be added at the end of the figure to explain what can be seen.
Author Response
Reviewer 1:
In their study, Zhang et al investigated the extent to which CVB3 can induce pyroptosis, a major mechanism leading to cell death, in colorectal carcinoma cells and in colorectal tumors in vivo. They demonstrated that CVB3 treatment induced cell death in vitro by pyroptosis via caspase-3-mediated cleavage of gasdermin E (GSDME). Moreover, CVB3-induced pyroptosis was promoted by reactive oxygen species (ROS). CVB3-induced pyroptosis in was confirmed in vivo after infection of colorectal tumors with CVB3.
The study is very well done and well written. Although it has already been shown that CVB3 can induce pyroptosis in CVB3-infected cells, this is the first time that this has been done in the context of oncolytic virotherapy in colorectal tumor cells. Thus, the focus of the paper and the main messages relate to elucidating the mechanism of pyroptosis occurring in colorectal tumor cells rather than the efficacy and safety or further development of CVB3 virotherapy.
Our response: We are very grateful to the reviewer for these insightful comments on our manuscript. The reviewer's suggestions have greatly improved the quality of our manuscript. In the revised version, we carefully revised the concerns raised by the reviewer. We hope that our revised version will satisfy the reviewer.
Main criticism
It would certainly have been better if the authors had focused on elucidating the mechanism of pyroptosis by CVB3 in colorectal cells and colorectal carcinomas rather than repeatedly focusing on CVB3 virotherapy. This is particularly concerning because the presentations on CVB3 virotherapy give a rather distorted picture of the current state of the art. For example, there are already a number of papers that explicitly address the treatment of tumors, including colorectal carcinomas, with oncolytic CVB3. These (for example, Hazini et al. 2018 and 2021) have shown that wild-type CVB3 causes severe side effects and that more advanced CVB3s (CVB3 mutants or microRNA-regulated CVB3s) are more suitable for cancer therapy than the wild-type used in this study. Unfortunately, these studies were not cited and the data generated by the authors were not presented in the context of the previous results. Another example from the discussion section should be mentioned. The authors mention that in the future they plan to investigate the potential of CVB3 in the treatment of metastatic colorectal cancer by inducing two tumors in nude mice, one of which is injected with a CVB3. These studies have already been done in the 2018 and 2021 papers by Hazini et al. The introduction and the discussion should therefore be carefully revised once again.
Our response: First of all, we would like to thank the reviewer for these pertinent comments and suggestions. We very much agree with the reviewer's suggestion. Our research is really focused on exploring that CVB3 induce pyroptosis in colon cancer cells, so as to achieve the purpose of tumor treatment. We are also very grateful to the reviewers for pointing out the incompleteness of our references. For these problems, we have made targeted modifications in the main text.
In view of the reviewer's concern that there have been studies using wild-type CVB3 or advanced CVB3 for the treatment of colon cancer, and our paper does not cite these relevant studies. We have made targeted modifications in the abstract, introduction and other sections, as follows: “Our findings indicate that CVB3 have oncolytic activity in colon cancer cell lines via GSDME-mediated pyroptosis”, at lines 30-31 of Abstract section in revised manuscript; “Hazini et al. have confirmed that both wild-type CVB3 (WT-CVB3) and genetically engineered CVB3 have potential as oncolytic viruses in the treatment of colon cancer [10,11]. However, their work focused more on the effectiveness of cancer therapy, while ignoring the exploration of the molecular mechanism with which WT-CVB3 kills colon cancer cells. Therefore, we sought to clarify the molecular mechanism of WT-CVB3-linked colon cancer cell death to better facilitate the clinical application of CVB3 as an oncolytic virus”, at lines 54-60 of Introduction sections in revised manuscript; “Our findings indicate that WT-CVB3 has a therapeutic effect against colon cancer via GSDME-mediated pyroptosis”, at lines 76-77 of Introduction section in revised manuscript.
In response to the second question that reviewer paid attention to, previous studies have evaluated the therapeutic effect of CVB3 on metastatic colon cancer. In the discussion section, we made targeted modifications to the contents of these prospects. The details are as follows: “The inhibition of colon cancer tumor metastasis by oncolytic viruses is one of the key indicators of their therapeutic effect [1]. After the oncolytic virus infects cancer cells, it changes the tumor microenvironment, stimulating the antiviral components of the immune system which kill tumor cells. Activation of systemic immunity inhibits tumor spread and prevents recurrence [30,35]. Previous studies have also revealed the high potential efficacy of WT-CVB3 and genetically engineered CVB3 against metastatic colon cancer [10,11]. However, the molecular mechanism by which CVB3 acts against metastatic colon cancer has not been fully elucidated. Therefore, in subsequent experiments, we will mimic the metastasis of colon cancer by performing xenografts on both sides of immunodeficient nude mice, injecting virus on only one side, determine the protein levels in signal pathways related to cell death, and measure tumor volume on the opposite side to check for potential effects on metastatic tumors and regulatory mechanisms”, at lines 384-395 of revised manuscript.
Minor points.
1 In Figure 2C, arrows should be added (as in 2B) to show the pyroptosis bubble.
Our response: We appreciate the reviewer’s suggestion, and we have modified the Figure 2C in revised version of manuscript.
2 In Figure 3C, the expression of GSDMD is shown in the Western blot. Several bands can be seen here. Which of the bands corresponds to GSDMD? Figure 3C also shows an increase in the second band from the top for GSDMD starting at 15 hours and a decrease in the top band for pro-caspase-1. The significance of the bands and the reason for the change is not clear to the reviewer and is not explained in the text but should be explained.
Our response: We are very sorry that the results and data we submitted may have misled the reviewers. In HT-29 cells, CVB3 infection does not activate GSDMD pathway and Caspase-1 expression. These problems mentioned by reviewer may be the result of non-specific reaction of antibody. We have re-tested these protein samples with new antibodies in the revised version. The results were presented in the newly revised manuscript. The results showed that infection of CVB3 could not activate GSDMD pathway and Caspase-1 expression after 24 hours in HT-29 cells. Protein levels were consistent at all time points. Once again, we apologize to the reviewer for our careless handling of data and experiments.
3 Figure 6F. The figure is much too small; even at maximum magnification on the computer, the differences are barely visible. A sentence should be added at the end of the figure to explain what can be seen.
Our response: We appreciate the reviewer for pointing this issue. In the revised version of manuscript, we not only adjusted the size of these immunohistochemical images for the convenience of reviewers and readers, but also added the description of this result in Figure Legend. The details are as follows: “WT-CVB3 injection increased the expression of GSDME and Caspase-3 in the tumor tissues of treated mice compared with that in those in the UV-CVB3 group”, at lines 320-322 of revised manuscript.

Reviewer 2 Report
In their manuscript "Coxsackie viruses Group B3 have oncolytic activity against colon cancer through gasdermin E-mediated pyroptosis", Zhang et al. analyse the oncolytic potential of wild-type CVB3 on colon cancer, both in vitro and in vivo. Their study links the oncolytic effect of CVB3 to the induction of pyroptosis in infected cells. This manuscript provides valuable insight into the mechanism of virus-induced pyroptosis. However, the following points should be addressed prior to publication.
1. 1) Please provide all necessary information in the Material and Methos section (e.g. concentration of used inhibitors Z-DEVD-FMK and Selonsertib, used flow cytometry channels, MOI used for in vitro infection experiments, antibody information (clone, dilution), injection of mice (i.v., s.c., or i.t.).
2. 2) Fig 1B: Please correct toAnnexin
3. 3) In general, the authors should provide a more detailed description of the observed data within the Results section. Concrete data (e.g. fold-reduction of viability) in the text would give valuable information to the reader and, therefore, significantly improve the manuscript.
4. 4) Fig1: Why did the authors choose different end-points for HT-29 and HCT-116 cells. Were MOI-dependent effects in the used cell lines determined time-dependent, and do the chosen end-points correlate with the duration of CVB3 replication-cycle?
5. 5) Lines 208-211: The authors conclude that neither casp-1 nor GSDMD was activated by CVB3 infection. This contradicts Fig. 3C/D, where a clear change in the band pattern of Pro-caspase 1 and GSDMD is visible. Please clarify.
6. 6) Fig. 4B: the Western Blot panel of cleaved caspase-3 is missing.
7. 7) Line 230: The authors state that "NAC inhibited casp-3 activation and GSDME cleavage to some extent", and further in the Discussion section (lines 329-330) that statistical analysis suggested that pretreatment with NAC inhibited pyroptosis. This effect can hardly be concluded by eye, and the stated statistical analyses suggest that these data were quantified. Please provide the quantification data.
8. 8) Is the oncolytic effect of CVB3 restricted to Colon cancer cells? Since the primary receptor for CVB3 cell entry is the CAR, it is very likely that not only colon cancer cells will be infected by CVB3, but basically every cell or tissue that expresses CAR. Moreover, OVs are usually equipped with a mechanism that restricts replication to tumor cells leaving healthy cells intact. Using the wt CVB3 is a major weakness of this manuscript since also healthy cells will be affected.
9. 9) In this study, the authors used an artificial mouse model that enables the application of CVB3 intratumoral. However, this application route is most likely not suitable for human patients, making an intravenous application necessary. This, in turn, is associated with many challenges and outlines at least the urgency of a tumour-restricted replication mechanism. E.g. demonstrating the efficient killing of tumour cells (vs. healthy cells) of a CVB3 equipped with a tumour-specific promoter would significantly strengthen this manuscript. Further, additional arming strategies for colon cancer should be discussed.
1 10) Pyroptosis is a double-edged sword concerning cancers, as the inflammatory situation can either promote or inhibit tumorigenesis. This should be discussed in the light of clinical therapies.
Author Response
Reviewer 2
In their manuscript "Coxsackie viruses Group B3 have oncolytic activity against colon cancer through gasdermin E-mediated pyroptosis", Zhang et al. analyse the oncolytic potential of wild-type CVB3 on colon cancer, both in vitro and in vivo. Their study links the oncolytic effect of CVB3 to the induction of pyroptosis in infected cells. This manuscript provides valuable insight into the mechanism of virus-induced pyroptosis. However, the following points should be addressed prior to publication.
Our response: We are very grateful to the reviewer for these insightful comments on our manuscript. The reviewer's suggestions have greatly improved the quality of our manuscript. In the revised version, we carefully revised the concerns raised by the reviewer. We hope that our revised version will satisfy the reviewer.
1) Please provide all necessary information in the Material and Methos section (e.g. concentration of used inhibitors Z-DEVD-FMK and Selonsertib, used flow cytometry channels, MOI used for in vitro infection experiments, antibody information (clone, dilution), injection of mice (i.v., s.c., or i.t.).
Our response: Thanks to the reviewer for these suggestions. Except that we have indicated the MOI of virus we used in the experiment in line 98, we have supplemented other information in the revised version of manuscript. See line 97, 105, 118, 137-140 and 149 of the revised manuscript for details. We hope that our revisions can enable readers to effectively follow our methods and results and satisfy reviewers.
2) Fig 1B: Please correct to Annexin
Our response: We appreciate the reviewer for pointing out this error, and we apologize for our carelessness in preparing these figures. These problems will be corrected in the revised manuscript.
3) In general, the authors should provide a more detailed description of the observed data within the Results section. Concrete data (e.g. fold-reduction of viability) in the text would give valuable information to the reader and, therefore, significantly improve the manuscript.
Our response: We are very grateful to the reviewer for this suggestions. This suggestion can greatly improve the readability of the manuscript and enable readers to accurately understand our results. Due to the overall length, we have added more concrete results description in the partial results section, such as: “Statistical analysis of the flow cytometry results revealed that more than 60% of the HT-29 cells died within 24 hours of CVB3 infection (Figure 1B). These results suggest that CVB3 kills the HT-29 colon cancer cell line. We observed a similar effect on HCT-116 cells, with more than 90% of them dying within 48 hours of CVB3 infection”, at lines 179-183; “Under fluorescence microscopy, we detected the DCFH-DA levels and observed a significant increase in ROS after infection with CVB3”, at lines 242-243; “This suggests that the treatment with WT-CVB3 may have mildly damaged normal tissue and cells in the mice and not have given mice a survival advantage”, at lines 299-301 of revised manuscript.
4) Fig1: Why did the authors choose different end-points for HT-29 and HCT-116 cells. Were MOI-dependent effects in the used cell lines determined time-dependent, and do the chosen end-points correlate with the duration of CVB3 replication-cycle?
Our response: This is a very good question. This is also the difference between the two different colon tumor cells infected by CVB3 in the actual experiment. We strongly agree with the reviewer that this difference may be related to the replication cycle of viruses in different cells. We chose different endpoints in the two cell lines to detect cell death, mainly due to the inconsistent replication cycle of CVB3 in the two cell lines. The proliferation rate of CVB3 in HT29 cells was higher than that in HCT116 cells. Therefore, in this experiment, we chose an end-point of 24 hours for HT29 cells infected with CVB3, while 48 hours for HCT116 cells, to observe the changes of cell status and characteristics after virus infection. In order to eliminate the misunderstanding of reviewers and readers, we added a description to explain why we chose different end-points in HCT116 and HT29 cells to observe the characteristics of cell death. The details are as follows: “CVB3’s replication cycles are inconsistent in HCT116 and HT29 cells, leading us to select different experimental end-points to observe the phenomenon of virus-induced cell death and perform other related experiments”, at lines 94-96 of revised manuscript.
5) Lines 208-211: The authors conclude that neither casp-1 nor GSDMD was activated by CVB3 infection. This contradicts Fig. 3C/D, where a clear change in the band pattern of Pro-caspase 1 and GSDMD is visible. Please clarify.
Our response: We are very grateful to the reviewer for the concern about our results. In HT-29 cells, it can be observed that Caspase-1/GSDMD signaling pathway will not be activated, either in the data of the previous version or in the data of the new version with non-specific bands removed (Figure 3C). However, the reviewer were more worried about the activation of Caspase-1/GSDMD signal pathway in HCT-116 cells at the 30th hour after virus infection. We repeated the experiment three times and got similar results. Therefore, we believe that CVB3 infection of HCT-116 cells will firstly induce GSDME-mediated pyroptosis at the 12th hour, and with the accumulation of metabolites, it will induce GSDMD-mediated pyroptosis at the 30th hour. By comparing the whole process, it can also be confirmed that CVB3 infection mainly mediates GSDME pathway leading to pyroptosis of colon cancer cells.
In order to fully let reviewer and readers understand this process, and to make the description of the results more rigorous, we have revised the description of the results. The details are as follows: “Neither casp-1 nor GSDMD were activated in HT-29 cells (Figure 3C and Figure S1). Although we observed the activation of the casp-1/GSDMD pathway in HCT116 cells 30 hours after CVB3 infection ( in comparison, GSDME-mediated pyroptosis occurred 12 hours after CVB3 infection), we believe that CVB3 infection of HCT116 causes pyroptosis mainly via the casp-3/GSDME pathway, rather than the casp-1/GSDMD pathway that subsequently occurred (Figure 3D) [18]. The statistical analysis results of densitometry analysis of the digital images of Western blots in Figure 3 are shown in Figure S1. These data suggest that CVB3-induced pyroptosis of colon cancer cell lines occurs via the casp-3/GSDME pathway, rather than the casp-1/GSDMD pathway”, at lines 220-229 of revised manuscript.
In addition, in the discussion section, we also added that we observed both casp-3/GSDME and casp-1/GSDMD pathways regulating the mechanism of pyroptosis in HCT-116 cells, and explained the possible mechanisms. The details are as follows: “We explored the molecular mechanism by which CVB3 infection leads to pyroptosis in two colon cancer cell lines. Of note, after CVB3 infects HCT-116 cells, the casp-3/GSDME pathway is activated at 12 hours post-inoculation, and the casp-1/GSDMD pathway is active at 30 hours. This indicates that the pyroptosis caused by CVB3 action in HCT-116 cells may be mediated by the casp-3/GSDME pathway first, and then the casp-1/GSDMD pathway can be activated by the action of cell metabolites and other substances. This phenomenon has also been reported in other studies, but the details of the regulatory mechanism need to be clarified with further research [25,26]. Based on the results with these two cell lines, we established that CVB3 infection is mainly mediated via the casp-3/GSDME pathway, leading to pyroptosis of colon tumor cells” at lines 335-345 of revised manuscript. We hope that this explanation will satisfy the reviewer.
6) Fig. 4B: the Western Blot panel of cleaved caspase-3 is missing.
Our response: Thank the reviewers for pointing out our mistakes. We are very sorry that we missed the immunoblot results of cleaved caspase-3 in Figure 4B when preparing data and figures. In the revised version of manuscript, we have added the corresponding data to Figure 4B of the revised version, and added the corresponding original data to the supplementary materials.
7) Line 230: The authors state that "NAC inhibited casp-3 activation and GSDME cleavage to some extent", and further in the Discussion section (lines 329-330) that statistical analysis suggested that pretreatment with NAC inhibited pyroptosis. This effect can hardly be concluded by eye, and the stated statistical analyses suggest that these data were quantified. Please provide the quantification data.
Our response: Yes, we very much agree with the reviewers, and we are very grateful to the reviewers for pointing out this problem. It is true that the expression levels of proteins cannot be clearly determined by naked eyes through the bands of immunoblotting. During the experiment, we repeated these experiments for three times, and analyzed the gray levels of the bands by synthesizing the data of the three times, and finally obtained the statistical quantitative analysis. In the revised version, we have performed band grayscale analysis and statistical analysis on all immunoblot results, and added the results in the form of supplementary figures. See the revised version for details. We hope that our revision can ensure the accuracy and scientificity of our data and satisfy the reviewer.
8) Is the oncolytic effect of CVB3 restricted to Colon cancer cells? Since the primary receptor for CVB3 cell entry is the CAR, it is very likely that not only colon cancer cells will be infected by CVB3, but basically every cell or tissue that expresses CAR. Moreover, OVs are usually equipped with a mechanism that restricts replication to tumor cells leaving healthy cells intact. Using the wt CVB3 is a major weakness of this manuscript since also healthy cells will be affected.
Our response: We agree with the reviewer's opinion. The oncolytic effect of CVB3 is indeed not limited to colon cancer cells, normal cells can also be infected by CVB3. In the discussion section, we also mentioned that CVB3 may cause some safety issues through intratumoral injection (lines 349-356). In order to better let the readers understand that we have considered the security of CVB3 in the manuscript, we have supplemented the following description in the revised version. The details are as follows: “This suggests that intratumorally-injected CVB3 may have spread, causing myocarditis or pancreatitis. This may have been because CVB3 can infect normal cells as well, which also express coxsackievirus and adenovirus receptor (CAR) [30]”, at lines 368-371 of revised manuscript. Moreover, in this work, we use intratumoral injection to evaluate the therapeutic effect of CVB3 on colon tumors, which is mediated by inducing Caspase-3/GSDME-mediated pyroptosis. In the answer to the next question, we have also supplemented the experiments to be carried out in the future, focusing on finding a balance between viral toxicity and therapeutic effect to better treat colon tumors. The specific modifications are explained in detail in 9).
9) In this study, the authors used an artificial mouse model that enables the application of CVB3 intratumoral. However, this application route is most likely not suitable for human patients, making an intravenous application necessary. This, in turn, is associated with many challenges and outlines at least the urgency of a tumour-restricted replication mechanism. E.g. demonstrating the efficient killing of tumour cells (vs. healthy cells) of a CVB3 equipped with a tumour-specific promoter would significantly strengthen this manuscript. Further, additional arming strategies for colon cancer should be discussed.
Our response: Thanks very much for the questions from the reviewer, and we agree with the advice that intratumoral CVB3 administrations are not suitable for humans and should be administered intravenously. However, as mentioned in the previous question, if WT-CVB3 is used intratumoral, diseases such as pancreatitis and myocarditis can be triggered in infants because not only colon cancer cells, but also other healthy cells will be infected with CVB3 (lines 368-371). However, the injection intravenously of WT-CVB3 will aggravate these conditions. Therefore, we make a xenograft model and used WT-CVB3 for intratumoral injection, mainly to study the mechanism of CVB3 oncolytic effect, and lay a good foundation for clinical research and application. In order to better let reviewers and readers understand why we choose intratumoral injection, we have made targeted modifications in the discussion, as follows: “Intratumoral CVB3 administration is less preferable than the intravenous injection of human patients in clinical trials. However, intravenous injection is more likely to aggravate the disease because CVB3 will infect normal cells and tissues. To prevent CVB3 from infecting healthy cells after entering the bloodstream, which can cause other diseases, WT-CVB3 should be modified to make it more suitable for clinical use. Its toxicity must be reduced and its safety and anti-tumor efficacy enhanced”, at lines 396-401 in revised manuscript.
At the same time, we are grateful to the reviewers for their suggestion of "a CVB3 equipped with a tumour-specific promoter", which greatly promoted our research progress. In the revised version, we have added a prospect for the next phase of experiments to ensure the safety and tumor targeting of CVB3. Therefore, we added “The "hypoxia" feature is common in tumor cells but not seen in normal cells. We may be able to engineer CVB3 with a tumor-specific hypoxia-inducible factor (HIF)-responsive promoter which targets infected tumor cells without infecting normal cells”, at lines 401-404 in revised manuscript. And added “In subsequent studies, we hope to combine the above improved methods to develop a truly effective, high-security oncolytic virus which can be used for clinical treatment”, at lines 398-400 in revised manuscript.
10) Pyroptosis is a double-edged sword concerning cancers, as the inflammatory situation can either promote or inhibit tumorigenesis. This should be discussed in the light of clinical therapies.
Our response: We would like to thank the reviewer for the pertinent suggestions. The inflammatory response caused by pyroptosis should be taken into account in subsequent studies. We have made modifications in the discussion, as follows: “Pyroptosis is a double-edged sword. It can open up new tumor treatment strategies by inducing cell death, but studies have also shown that excessive pyroptosis may also cause a serious inflammatory response, damage normal tissues, and lead to inflammatory diseases [42]. We confirmed that CVB3 infection will lead to colon tumor cell death by mediating pyroptosis but it can also reduce the weight of mice. Therefore, in the subsequent clinical application of CVB3-mediated pyroptosis for colon tumor treatment, it is necessary to explore the dynamic balance between tumor cell death caused by pyroptosis and the inflammatory response induced by this process as well as considering tumor cell targeting and the safety of viruses”, at lines 417-425 in revised manuscript.

Round 2
Reviewer 2 Report
Thank you very much for allowing me to review the revised version of this manuscript. In the revised manuscript, Zhang et al. carefully and satisfactorily addressed all points made by the reviewer. As such, they included additional explanations and comprehensively extended the discussion section. Therefore, I have no further concerns regarding the publication of this manuscript.